# Effect of Inhalation Profile on Delivery of Treprostinil Palmitil Inhalation Powder

**DOI:** 10.3390/pharmaceutics15030934

**Published:** 2023-03-14

**Authors:** Helena Gauani, Thomas Baker, Zhili Li, Vladimir S. Malinin, Walter R. Perkins, Eugene J. Sullivan, David Cipolla

**Affiliations:** Insmed Incorporated, Bridgewater, NJ 08807, USA

**Keywords:** device performance, dry powder inhaler (DPI), fine particle dose (FPD), aerosol delivery, inhalation, treprostinil palmitil (TP), treprostinil palmitil inhalation powder (TPIP), pulmonary arterial hypertension (PAH), interstitial lung disease (ILD)

## Abstract

Treprostinil palmitil (TP), a prodrug of treprostinil, is being developed as an inhalation powder (TPIP) for the treatment of patients with pulmonary arterial hypertension (PAH) and pulmonary hypertension due to interstitial lung disease (PH-ILD). In ongoing human clinical trials, TPIP is administered via a commercially available high resistance (HR) RS01 capsule-based dry powder inhaler (DPI) device manufactured by Berry Global (formerly Plastiape), which utilizes the patient’s inspiratory flow to provide the required energy to deagglomerate and disperse the powder for delivery to their lungs. In this study, we characterized the aerosol performance of TPIP in response to changes in inhalation profiles to model more realistic use scenarios, i.e., for reduced inspiratory volumes and with inhalation acceleration rates that differ from those described in the compendia. The emitted dose of TP for all combinations of inhalation profiles and volumes ranged narrowly between 79 and 89% for the 16 and 32 mg TPIP capsules at the 60 LPM inspiratory flow rate but was reduced to 72–76% for the 16 mg TPIP capsule under the scenarios at the 30 LPM peak inspiratory flow rate. There were no meaningful differences in the fine particle dose (FPD) at all conditions at 60 LPM with the 4 L inhalation volume. The FPD values for the 16 mg TPIP capsule ranged narrowly between 60 and 65% of the loaded dose for all inhalation ramp rates with a 4 L volume and at both extremes of ramp rates for inhalation volumes down to 1 L, while the FPD values for the 32 mg TPIP capsule ranged between 53 and 65% of the loaded dose for all inhalation ramp rates with a 4 L volume and at both extremes of ramp rates for inhalation volumes down to 1 L for the 60 LPM flow rate. At the 30 LPM peak flow rate, the FPD values for the 16 mg TPIP capsule ranged narrowly between 54 and 58% of the loaded dose at both extremes of the ramp rates for inhalation volumes down to 1 L. Based on these in vitro findings, the TPIP delivery system appears not to be affected by the changes in inspiratory flow profiles or inspiratory volumes that might be expected to occur in patients with PAH or PH associated with underlying lung conditions such as ILD.

## 1. Introduction

Inhaled treprostinil (TRE), an approved prostacyclin analog pulmonary vasodilator treatment for PAH patients, is available as a solution for inhalation (Tyvaso^®^, United Therapeutics) [1] and as a dry powder formulation containing either 16, 32, 48, or 64 µg TRE (Tyvaso DPI^®^, United Therapeutics) [2]. TRE requires frequent daily dosing (labeled for administration at four separate events daily) to provide sustained pulmonary vasodilation [1,2] due to its rapid absorption from the lung and short systemic half-life. Adverse local and systemic events such as cough, headache, and throat irritation have been widely reported [1,2,3].

A prodrug of TRE, treprostinil palmitil (TP), has been developed to potentially improve the duration of efficacy and minimize both the local side effects and those related to peak systemic exposure. TP is a covalently bonded 16-carbon alkyl chain through an ester bond to TRE, and in its suspension formulation (TPIS), pressurized metered dose inhaler formulation (TPIA), and dry powder formulation (TPIP), the persistent formation of TRE is generated through the action of endogenous lung esterases; it has been shown to be efficacious for up to 24 h post-administration in rat and dog pK studies [4,5,6,7]. Furthermore, in guinea pig studies designed to evaluate the provocation of cough by inhaled TP versus TRE, coughing was observed only at significantly higher TPIP doses compared to TRE, which may be attributed to the slow conversion rate of the prodrug to TRE in the lung [8,9].

In its dry powder form, TPIP is generated by a spray drying process with mannitol and leucine as excipients. Recent in vitro studies evaluating the performance of TPIP in multiple DPIs resulted in the selection of the HR RS01 device as it demonstrated consistent aerosol performance at 40, 60, and 80 LPM flow rates using the compendial-recommended rectangle-wave inspiratory profile and a 4 L inspiratory volume [10]. These flow rates were chosen to span a range of possible inspiratory efforts that subjects may apply to the device: 5.4 kPa pressure drop for 80 LPM, 4.0 kPa for 60 LPM, and 1.4 kPa pressure drop for 40 LPM [10]. The in vitro performance of the device as assessed by fine particle dose (FPD) for capsules containing 8, 16, and 32 mg TPIP was independent of the inspiratory flow rate, suggesting that the pressure drop across the device required to fully deagglomerate and disperse the TPIP powder is no more than 1.4 kPa, which is likely achievable even by subjects with significantly impaired pulmonary function.

We further evaluated the in vitro aerosol performance of capsules containing 16 and 32 mg TPIP utilizing the HR RS01 device in response to more clinically relevant inhalation profiles deviating from the rectangular wave profile. The aerosol performance of the 16 mg TPIP capsule was also evaluated at reduced inhalation volumes of 2 and 1 L and at a reduced peak inspiratory flow rate of 30 LPM, corresponding to a pressure drop of 0.89 kPa across the device to reflect a more extreme set of user scenarios.

In the first set of experiments, we retained the 4 L total inhalation volume, but instead of an instantaneous transition from 0 to 60 LPM as defined in the compendia (i.e., the rectangle wave profile), we introduced variable inhalation ramp rates to delay the time to reach the nominal inspiratory flow rate of 60 LPM: inhalation acceleration rates of 20, 5.88, 2, 1, and 0.5 L/s^2^ were used, corresponding to ramp times of 50, 170, 500, 1000, and 2000 ms, respectively. In the second set of experiments, we repeated the performance characterization at both extremes of ramp times, 50 and 2000 ms, using smaller inhalation volumes of 2 L and 1 L to simulate use by a subject with compromised lung volumes. In the third set of experiments, the performance characterization was repeated at both extremes of acceleration rates, 20 and 0.5 L/s^2^, to achieve the reduced inspiratory flow rate of 30 LPM. Thus, the resulting data may inform about device performance in TPIP clinical trials in patients with respiratory muscle dysfunction or compromised lung volumes.

## 2. Materials and Methods

### 2.1. DPI Device

An RS01 Model 7 (Berry Global, Osnago, Italy) high resistance (HR) (code 239700002AA) DPI device was used for evaluation. The pressure drop across the HR device containing an empty capsule was measured as a function of the flow rate between 30 and 100 L/min (LPM) (Appendix A), as reported previously [10].

### 2.2. Compounds and Materials

We have previously described the GLP production scale manufacture of TPIP by spray drying at Bend Research Inc. (Bend, OR, USA) and its characterization [10]. Because TP is relatively potent, TPIP powder contains only 1.0% TP, with the rest of the powder comprised of mannitol and leucine excipients in a 70 to 30 ratio by weight. TPIP is a fine powder with a D50 of 1.7 µm. After the manufacture of TPIP, 16 or 32 mg is filled into size #3 capsules (Vcaps^®^ Plus Hypromellose, Capsugel Manufacturing Inc., Greenwood, SC, USA), representing a loaded dose of 160 or 320 µg TP, respectively.

### 2.3. Aerosol Characterization Using the Next Generation Impactor (NGI) and Breathing Simulator

To evaluate the differences in aerosol performance in response to different breathing profiles, TPIP aerosol emitted from the HR device was delivered to the NGI (MSP Corp., Shoreview, MN, USA) at ambient temperatures and low humidity (<45% RH) under various scenarios generated by the breathing simulator (BRS 3100, Copley Scientific Ltd., Nottingham, UK) (Appendix A). A mixing inlet (Copley) was attached in between the pre-separator and the induction port to allow a constant flow rate of 30 or 60 LPM to be achieved through the NGI while varying the change in inhalation profiles from the HR device via the BRS system. The outlet port of the NGI was connected to a 47 mm glass fiber filter (Type A/E, Pall Life Sciences, Port Washington, NY, USA) to collect the aerosol fines that escape deposition in the NGI, followed by a vacuum pump drawing 30 or 60 LPM airflow (X) via a critical flow controller (TPK 2100, Copley). An airflow of 30 or 60 LPM (Y) was drawn into the mixing inlet to ensure that initially no airflow (X-Y) was drawn through the inhaler.

Aerosolization of the powder into the NGI was triggered by the BRS system via a diversion valve that redirects the airflow (Y) into the BRS system. The BRS software was manually programmed through a Word Pad script to simulate inhalation acceleration rates of 20, 5.88, 2, 1, and 0.5 L/s^2^ corresponding to 50, 170, 500, 1000, and 2000 millisecond (ms) ramp times for a total inhaled volume of 4 L, and for 2 and 1 L inhalation volumes at both extreme acceleration rates to achieve the 60 LPM flow rate (Appendix A). Additionally, the breathing profile scripts for the 1, 2, and 4 L inhalation volumes at 20 and 0.5 L/s^2^ (corresponding to 25 and 1000 ms ramp times) to reach the 30 LPM flow rate are included. A trapezoidal waveform was used to maintain the 4 L total inhalation volume as well as to keep the maximum inhalation flow rate at 60 LPM, corresponding to the 4 kPa pressure drop across the device (Figure 1A). Additionally, the breathing profiles with reduced inhalation volumes of 1 and 2 L were evaluated at the two extreme acceleration rates of 20 and 0.5 L/s^2^ (corresponding to 50 and 2000 ms ramp times) for the 60 LPM inspiratory flow rate (Figure 1B), and 25 and 1000 ms ramp times for the 30 LPM inspiratory flow rate (Figure 1C).

For each experiment conducted at a specific inhalation profile setting, the aerosol performance of the TPIP drug product was characterized in terms of its emitted dose (ED) and particle size distribution (PSD), which included its mass median aerodynamic diameter (MMAD) and geometric standard deviation (GSD). Additionally, the fine particle fraction < 5 µm (FPF) and extra fine particle fraction < 2 µm (eFPF) and fine particle dose < 5 µm (FPD) and extra fine particle dose < 2 µm (eFPD) were also calculated. Finally, to ensure that each experiment was conducted properly, the total recovery of TP was measured and compared to the nominal loaded dose of TP in the capsule.

### 2.4. Quantitation of TP

TP was quantified by reverse phase HPLC using a mass spectrometry detector, as described previously [10]. The calibration range was 4–500 ng/mL of TP.

### 2.5. Determination of ED, MMAD, GSD, FPF and FPD

Total recovery of TP was expressed relative to the nominal loaded dose of TP in the capsule, which included quantitation of the residual powder in the capsule and the recovered mass of TP from all the components of the NGI and from the DPI device. The ED, MMAD, and GSD were calculated as noted previously [10] and in concordance with the compendia [11,12]. The ED represents the mass of TP recovered from the NGI and is expressed as a percentage of the nominal loaded dose of TP in the capsule.

A linear regression analysis was performed based on the normal probability z-score of the cumulative TP mass recovered (inverse of the standard normal cumulative distribution) versus the log cut-off diameters associated with NGI stages 1 to 7 at 60 or 30 LPM flow rates. The cut-off diameters for NGI stages 1 to 7 were 8.06, 4.46, 2.82, 1.66, 0.94, 0.55, and 0.34 µm at the 60 LPM flow rate, respectively, and 11.72, 6.40, 3.99, 2.30, 1.36, 0.83, and 0.54 µm at the 30 LPM flow rate, respectively. The MMAD was calculated based on the normal probability z-score corresponding to the cumulative TP mass percentage of 50% and from the slope and intercept of the linear regression analysis from the NGI stages.

The FPF represents the proportion of TP that is present in particles with an aerodynamic diameter of less than 5 µm. It is reported both as a percentage of the emitted dose and the nominal loaded dose, but in the text, we discuss only the FPF as a percentage of the emitted dose. The eFPF represents the proportion of TP that is present in particles with an aerodynamic diameter of less than 2 µm. It is expressed both as a percentage of the emitted dose and the nominal loaded dose, but in the text, we discuss only the eFPF as a percentage of the emitted dose. The FPF (and eFPF) values are obtained from a normal probability z-score based on the log of the aerodynamic diameter of 5 µm (or 2 µm) and the slope and intercept of the linear regression analysis from the NGI stages. The FPD and eFPD are calculated from the FPF and eFPF as the mass of TP emitted dose with an aerodynamic diameter of less than 5 µm and 2 µm, respectively.

### 2.6. Data Analyses and Statistics

Mean values of ED, FPF, FPD, and eFPD were calculated for each of the inhalation profiles for TPIP capsule doses of 16 mg or 32 mg, containing 160 or 320 µg TP, respectively. A 2-tailed *t*-test analysis was used to compare individual results within the same group, and an ANOVA single-factor analysis was used for an overall comparison of results within a group in Microsoft Excel. Statistical significance was noted for *p*-values < 0.05, while non-significance was noted for *p*-values > 0.05. The best-fit linear regression analysis of the mean square root of the pressure drop as a function of the flow rate for the high-resistance device is presented in Appendix A. Additionally, the best-fit linear regression analysis was applied for the FPD trend response as a function of inhalation acceleration rate for each capsule tested.

## 3. Results

### 3.1. Evaluation of the Inhalation Acceleration Rates for the 16 and 32 mg TPIP Capsule Doses at 60 LPM Inspiratory Flow Rate and 4 L Inhalation Volume

The deposition profiles based on TP recovery from the NGI are shown in Figure 2A,B for the 16 mg and 32 mg TPIP capsules, respectively, using the 50, 170, 500, 1000, and 2000 ms ramp times to achieve the 60 LPM inspiratory flow rate with a 4 L inhalation volume. The TP aerosol particle size distributions for both capsule doses were polydisperse, but there was a small shift in distribution as a function of inhalation ramp times. As the inhalation ramp time increased, there was slightly greater deposition in stages 6 to the filter and slightly less in stages 2 to 5, with the greatest change for stage 3 and the filter.

The values for the ED, MMAD, GSD, FPF, and FPD for the 50, 170, 500, 1000, and 2000 ms ramp times for the 16 and 32 mg TPIP capsules are shown in Table 1. Additionally, the total mass recoveries and eFPF and eFPD are incorporated in Appendix A. The mean total recovery of TP ranged between 90 and 99% for the five different ramp conditions covering both TPIP capsule doses. This indicates that the experimental procedure was well executed.

The ED was independent of the capsule dose, ranging from 80 to 82% for the 160 µg TP in the 16 mg TPIP capsule and 81 to 89% for the 320 µg TP in the 32 mg TPIP capsule. Variations in inhalation ramp times did not alter the near-complete release of TPIP from the capsule compared to the compendial rectangular waveform condition.

The MMAD values increased with capsule load but remained within the respirable range (<5 µm), varying from 1.0 µm to 1.3 µm for the 16 mg TPIP capsule and increasing to 1.3 to 1.7 µm for the 32 mg TPIP capsule. The MMAD values were slightly higher for the shortest ramp times (50 ms and 170 ms) by 0.1 to 0.2 µm. This result suggests that a slower acceleration (i.e., a longer ramp time) to the 60 LPM flow rate was more effective at fully dispersing the TPIP powder. The GSDs were all above 3.5 for both doses, indicating significant polydispersity in aerosol particle size.

The FPF and FPD values were uniform across all five ramp times for the 16 mg TPIP capsule, ranging from 78 to 80% and 100 to 104 µg TP, respectively (Table 1 and Figure 2A). There were no statistical differences (*p* > 0.05) among the individual FPD results or overall for this group (Table 2). For the 32 mg TPIP capsule, the FPF and FPD values ranged from 68 to 73% and 176 to 209 µg TP, respectively, with the longer ramp times (i.e., slower acceleration rates) improving powder dispersion for the higher capsule load (Figure 2B). This led to a statistical difference (*p* < 0.05) in FPD for the 1000 ms ramp time compared to the 50 ms ramp time and for the 2000 ms ramp time compared to the 50, 170, and 500 ms ramp times, as well as for the overall FPD results for this group (*p* = 0.005) (Table 2). While there were statistical differences in FPD for some of the conditions, the plot of FPD as a function of inhalation acceleration rate for both capsule doses (Figure 3) shows that there were no meaningful differences in FPD at all conditions relative to traditional compendial delivered dose uniformity testing limits (75 to 125% of target) for DPIs [11].

The eFPF and eFPD were also uniform across all five ramp times for the 16 mg TPIP capsule, ranging from 58 to 63% and 74 to 81 µg TP, respectively (Appendix A). For the 32 mg TPIP capsule, the eFPF and eFPD values ranged from 48 to 54% and 125 to 156 µg TP, respectively, with the longer ramp times (i.e., slower acceleration rates) improving powder dispersion for the higher capsule load.

### 3.2. Evaluation of the Inhalation Volume for the 16 and 32 mg TPIP Capsule Doses at 60 LPM Inspiratory Flow Rate

Reduced inhalation volumes of 1 and 2 L for the 16 and 32 mg doses were evaluated at both the minimum (20 L/s^2^) and maximum (0.5 L/s^2^) inhalation acceleration rates to achieve the 60 LPM target. The deposition profiles based on TP recovery from the NGI as a function of inhalation volume for the 50 ms and 2000 ms ramp times are shown in Figure 4A for the 16 mg TPIP capsule and in Figure 4B for the 32 mg TPIP capsule. The TP aerosol particle size distribution profiles remained polydisperse across all inhalation volumes for the 16 mg and 32 mg TPIP capsules at both extremes of the inhalation flow rate ramp times to achieve the 60 LPM inspiratory flow rate, similar to what was observed for the acceleration study.

We summarized the ED, MMAD, GSD, FPF, and FPD (Table 3). Additionally, the total mass recoveries and eFPF and eFPD are incorporated in Appendix A. The mean total recovery of TP ranged between 90 and 99% for each of the experimental conditions. These results confirm that the experiments were executed properly.

The ED was independent of the dose, ranging from 79 to 83% for the 160 µg TP dose and 81 to 89% for the 320 µg TP dose. Reductions in inhalation volume down to 1 L did not impair the near-complete release of TPIP from the capsule.

The MMAD values increased slightly with capsule load but remained well within the respirable range (<5 µm), increasing from 1.1 µm to 1.3 µm for the 16 mg TPIP capsule to 1.3 to 1.7 µm for the 32 mg TPIP capsule. Decreasing the inhalation volume at both extremes of ramp times did not significantly affect the median aerosol particle size of TPIP. However, the MMAD values for the 16 mg and 32 mg TPIP capsules were highest for the fastest ramp time of 50 ms, similar to what was observed in the acceleration study, suggesting that the slower acceleration (i.e., a longer ramp time) was more effective at deagglomerating the TPIP powder. The GSDs were all above 3.5 for both doses, indicating significant polydispersity in aerosol particle size.

The FPF and FPD values were uniform across all inhalation volumes (1, 2, and 4 L) at both extremes of ramp time for the 16 mg TPIP capsule, ranging from 76 to 80% and 97 to 104 µg TP, respectively (Table 3 and Figure 5A). There were no significant differences among the individual FPD values and for the overall FPD results (Table 4) at the same acceleration rate, but individual differences were observed when comparing two extreme ramp times at the same inhalation volume for 50 and 2000 ms at 2 L, and when comparing two extreme ramp times with different inhalation volumes for 50 ms at 1 L and 2000 ms at 4 L, and for 50 ms at 2 L and 2000 ms at 1 L and 4 L. For the 32 mg TPIP capsule, the FPF and FPD values ranged from 63 to 68% and 169 to 178 µg TP for the 50 ms ramp time, respectively, and from 72 to 73% and 187 to 209 µg TP for the 2000 ms ramp time, respectively (Table 3 and Figure 5B). Significant differences between individual values were also observed when comparing the two extreme ramp times at the same inhalation volume, but these observations were consistent with those noted in the acceleration study for this capsule dose, and when comparing two extreme ramp times with different inhalation volumes. The longer ramp time resulted in a greater amount of TP being delivered in the respirable range (i.e., increased FPD) (Table 4).

### 3.3. Evaluation of the Inhalation Volume for the 16 mg TPIP Capsule Doss at 30 LPM Inspiratory Flow Rate

Reduced inhalation volumes of 1 and 2 L were evaluated for the 16 mg TPIP capsule dose at both the minimum (20 L/s^2^) and maximum (0.5 L/s^2^) inhalation acceleration rates to achieve the 30 LPM target. The deposition profiles based on TP recovery from the NGI as a function of inhalation volume for the 25 ms and 1000 ms ramp times are shown in Figure 4C. The TP aerosol particle size distribution profiles remained polydisperse across all inhalation volumes for the 16 mg TPIP capsule at both extremes of the inhalation flow rate ramp times to achieve the 30 LPM inspiratory flow rate. We summarized the total recovery for ED, MMAD, GSD, FPF, and FPD (Table 5). The mean total recovery of TP ranged between 95 and 101% for each of the experimental conditions. These results confirm that the experiments were executed properly.

The ED range was reduced to 72 to 76% for the 30 LPM flow rate as compared to the 79 to 83% range for the 60 LPM flow rate. Reductions in inhalation volume down to 1 L did not impair the release of TPIP from the capsule, but there was increased deposition in the RS01 device at the 30 LPM flow rate (17 to 25% TP) as shown in Figure 4C compared to the 60 LPM flow rate (10 to 12%) as shown in Figure 4A, indicating that the 0.89 kPa pressure drop generated at 30 LPM had insufficient energy to completely minimize deposition in the device.

The MMAD values remained well within the respirable range (<5 µm) from 1.9 µm to 2.3 µm but were higher than those observed for the same 16 mg TPIP capsule dose at the 60 LPM flow rate. Decreasing the inhalation volume at both extremes of ramp times did not significantly affect the median aerosol particle size of TPIP. However, the MMAD values were highest for the fastest ramp time of 25 ms, similar to what was observed in the inhalation volume study at the 60 LPM flow rate, suggesting that the slower acceleration rate (i.e., longer ramp time) may be more effective at deagglomerating the TPIP powder even at the very low flow rate of 30 LPM. The GSDs were all above 2.3, indicating significant polydispersity in aerosol particle size.

The FPF and FPD values were uniform across all inhalation volumes (1, 2, and 4 L) at both extremes of ramp times for the 16 mg TPIP capsule, ranging from 74 to 76% and 86 to 93 µg TP, respectively (Table 5 and Figure 5C). There were no significant differences among the individual FPD values and for the overall FPD results (Table 6) at the same acceleration rate, but individual differences were observed when comparing the two extreme ramp times at the same inhalation volume (i.e., for the 25 and 1000 ms ramp times at the 1 and 2 L volumes) and when comparing the two extreme ramp times at different inhalation volumes, consistent with that observed in the 60 LPM volume study for the same 16 mg TPIP capsule dose. However, for this study at 30 LPM, the longer ramp time resulted in less TP being delivered in the respirable range (i.e., decreased FPD) (Table 6), perhaps due to the lower energy required to disperse the powder at the reduced flow rate. While these differences in FPF and FPD were statistically significant, the absolute differences are not likely to be meaningful.

## 4. Discussion

TP was initially formulated as an inhaled suspension (TPIS) for nebulized delivery, as that provided the most rapid development path for confirmation that TPIS had a pharmacokinetic profile consistent with sustained conversion of TP to TRE [10]. However, to provide improved patient convenience, development efforts were conducted in parallel to formulate TP as both an inhaled MDI product [7] and a DPI product (TPIP). TPIP delivered in a DPI format was selected over the MDI product for subsequent clinical investigation in phase 1 dose escalation studies and is currently being evaluated in phase 2 studies in PH-ILD and PAH.

In general, effective drug aerosol performance for dry powder formulations is dependent on the combination of the device characteristics and formulation properties. Passive DPIs require an inspiratory flow rate through the device with adequate energy to deagglomerate and disperse the powder to reach the lungs, and this minimum force will vary depending upon the specific characteristics of that powder in that device. Regulatory compendia assume an inspiratory effort equivalent to a 4 kPa pressure drop [11,12], which may not be achievable by patients with respiratory muscle weakness or mechanical disadvantage. The high-resistance (HR) RS01 device was selected over other RS01 devices with different resistances based on the generation of a superior FPD with low variability along with dose proportionality from three different capsule doses (8, 16, and 32 mg TPIP) [10]. Subsequent in vitro studies were also performed to evaluate the effect of variations in inspiratory pressure on device performance [10]. The results of those studies demonstrated consistent aerosol performance independent of the TPIP capsule dose for inspiratory device flow rates both below (40 LPM flow rate corresponding to a 1.4 kPa pressure drop) and above (80 LPM corresponding to a 5.4 kPa pressure drop) the compendial 4 kPa pressure drop equivalent to a 60 LPM inspiratory flow rate [10]. Thus, these in vitro efforts were designed to extend those findings to model TPIP performance using more realistic inhalation profiles at both the compendial and non-compendial settings, and for subjects with reduced lung capacity down to a volume of 1 L to better simulate clinical use of the device by patients with impaired pulmonary function due to underlying disease.

In the present studies, various inhalation profiles were generated in vitro using the breathing simulator to model differences in the rate of ramp from 0 to 60 LPM inspiratory flow, keeping the total inspired volume constant at 4 L. The NGI was integrated into the breathing simulator to assess the impact on aerosol performance for inhalation profiles that differed from compendial NGI operation utilizing a rectangular wave profile utilizing a forceful and rapid, almost-instantaneous rate of acceleration to the target 60 LPM inspiration flow. We observed that variations in the inhalation acceleration rates using the HR RS01 device, ranging from a short ramp time of 50 ms to a long ramp time of 2000 ms (Table 1), did not compromise aerosol performance, with FPDs ranging from 100 to 104 µg TP for the 16 mg TPIP capsule load and 176 to 209 µg TP for the 32 mg TPIP capsule load, consistent with the previous in vitro HR device evaluation studies with TPIP under compendial NGI conditions: 104 and 197 µg TP for the 16 and 32 mg capsule loads, respectively [10]. A slight increase in the FPDs and a slight decrease in the MMADs were observed as the ramp time was lengthened, especially for the 32 mg TPIP capsule (Table 1). Thus, these results suggest that a more clinically relevant inhalation profile with a slower rate of acceleration in inhalation flow rate to the 60 LPM target, with its concomitant longer ramp time, may lead to slightly more favorable powder dispersibility and a greater FPD than that observed under the compendial testing conditions.

When developing a DPI product, one must understand the capability of subjects to use the device, and that assessment is influenced by their age, gender, lung condition, and their cognitive and physical capability to perform the appropriate breathing technique. With age, there are anatomical, physiological, and immunological changes that occur [13,14,15], ranging from decreased respiratory muscle strength to elasticity, which may affect the inhaled drug from effectively reaching the deep lungs. With gender differences, females have a 10–12% smaller lung volume than males of the same height and age [16]; for both sexes, taller individuals have a larger vital lung capacity [17,18]. With certain respiratory diseases such as COPD with severe airflow obstruction and hyperinflation, respiratory muscle function may be significantly impaired [19]. Therefore, clinicians must consider these factors when recommending the appropriate inhaled product for the patient [20].

To use passive DPIs correctly, the package insert typically instructs the subject to inhale rapidly and continue to inhale for as long as possible, to increase the likelihood that the full dose will be expelled from the capsule [21]. For the high-dose TOBI Podhaler (112 mg tobramycin, 28 mg per capsule for each of four capsules), a repeat inhalation may be required if powder remains in the capsule to ensure complete deaggregation and dispersion of the powder from the capsule [21]. For the TPIP product, we assessed whether in vitro device performance would be affected under conditions replicating inspiration scenarios for subjects with reduced lung volumes and respiratory muscle weakness. A reduction in the volume and internal pressure drop of inhalation air passing through the DPI may lead to an inability to disperse or deagglomerate all of the powder. Thus, inhalation volumes of both 2 and 1 L were evaluated for the dispersion of the 16 and 32 mg TPIP capsules in the HR RS01 device with both fast (20 L/s^2^) and slow (0.5 L/s^2^) acceleration rates to achieve the 60 LPM compendial target flow rate. Additionally, evaluation at these inhalation volume parameters at the reduced flow rate of 30 LPM was extended for the 16 mg TPIP capsule to assess the effects of a more reduced pressure drop generated during inhalation.

The device emitted doses (>79%) for the 1 and 2 L inhalation volumes that were comparable to those for the 4 L volume (Table 3) for the 60 LPM flow rate for both capsule doses, suggesting that the TPIP powder is nearly completely dispersed and exits the capsule within the first 1 L of inhaled volume. The MMAD, FPF, eFPF, FPD, and eFPD values were also similar to those for the 4 L volume under the same acceleration conditions (50 and 2000 ms ramp times). While statistically significant differences in FPD results were observed for some conditions with the 32 mg TPIP capsule, they do not appear to be clinically meaningful differences as they fall well within traditional compendial delivered dose uniformity requirements for DPIs [11]. The slower inhalation acceleration rate (2000 ms ramp time) resulted in improved FPDs over the inhalation volume range.

For the 30 LPM flow rate at the 1, 2, and 4 L inhalation volumes, emitted doses were reduced slightly to 72–75% at the slowest acceleration rate (1000 ms ramp time) compared to the fastest acceleration rate (25 ms ramp time) with an ED of 76% (Table 5). The 1 L inhalation volume did not impair the powder dispersion at either the fastest acceleration rate, with an FPD of 93 µg compared to 92 µg for the 4 L volume, or at the slowest acceleration rate, with an FPD of 87 µg compared to 90 µg for the 4 L volume. The MMAD and FPF values were also similar at the lower inhaled volume to those at 4 L but were modestly different from what was observed for the 60 LPM conditions. The increase in MMAD values (1.9 to 2.3 µm) and decrease in FPD values (86 to 93 µg TP) compared to the 60 LPM MMAD values (1.1 to 1.3 µm) and FPD values (97 to 104 µg TP) suggests slightly lower deagglomeration capabilities under extremely low DPI device pressure drop and inhalation rate conditions, but again, these differences do not appear to be clinically meaningful [11].

There are some limitations to this device performance study. The experiments were performed in vitro, and while modeling different inhalation profiles using a breathing simulator can replicate various inhalation profiles, these studies do not replicate all possible profiles. It is likely that inhalation profiles by real users will not conform to the profiles modeled here, which may lead to differences in the extent of powder extraction from the capsule (i.e., ED) or deaggregation (i.e., FPF). However, aerosol performance studies investigating a spray-dried tobramycin powder using a medium-resistance RS01 device suggested that capsule emptying upon dynamic inhalation of 1 to 1.5 L by healthy volunteers was comparable to the in vitro ED measurements at the compendial 4 kPa pressure drop setting, and superior to those at reduced inspiratory pressures [22]. These results suggest that our in vitro modeling may not be an overestimate of device performance in vivo.

The particle size distribution of TPIP is defined by a smaller MMAD than for nebulized TRE and a greater amount of TP in the extra-fine particle size fraction (<2 µm). A smaller particle size may lead to more peripheral lung deposition of TP [23]. While there is limited literature evidence discussing the optimal regional site of lung deposition for prostanoids, including TRE, the extended residence time of TP in the lung may lead to some redistribution of TP more centrally due to mucociliary clearance prior to particle dissolution and prodrug conversion to TRE [23]. Thus, a smaller particle size distribution may be desirable for a prodrug with longer retention in the lung prior to conversion to the active moiety.

Finally, in vitro TPIP device performance studies may not translate to efficacy in the clinic. Clinical trials, which are currently underway in PAH and PH-ILD, will be essential for establishing the in vivo safety and efficacy of TPIP.

## 5. Conclusions

These results confirm that the performance of TPIP in the HR RS01 device is relatively insensitive to changes in inhalation profiles with varied ramp rates to achieve the 60 LPM flow rate for the compendial volume of 4 L down to a reduced volume of 1 L. Minimal changes in aerosol performance were observed in response to the inhalation profiles generated from either slow or fast ramp rates, a reduction in target flow from 60 to 30 LPM flow rate, or reductions in inhalation volumes down to 1 L. These in vitro results suggest that subjects with significant respiratory muscle dysfunction or compromised lung capacity should be able to utilize the TPIP device without a meaningful deterioration in device performance or reduction in lung dose of TP.

## Figures and Tables

**Figure 1 pharmaceutics-15-00934-f001:**
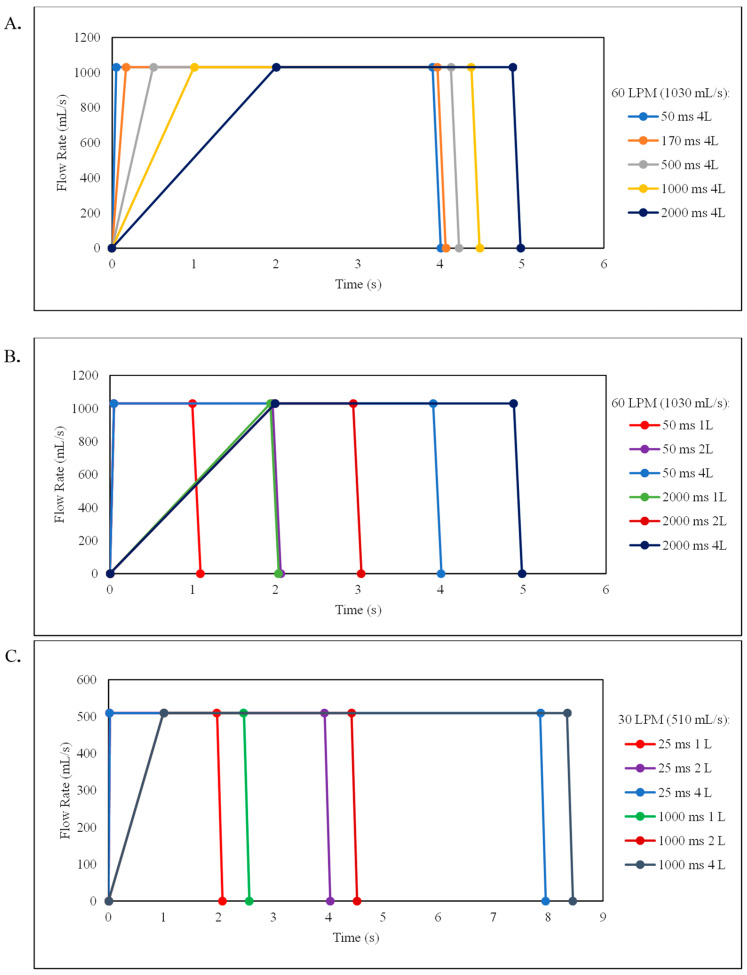
Trapezoidal waveforms for various inhalation profiles at peak inspiratory flow rates of 60 LPM (1030 mL/s) and 30 LPM (510 mL/s) associated with data from Appendix A. (**A**) The profiles show changes in acceleration rates: 50 ms (blue), 170 ms (orange), 500 ms (gray), 1000 ms (yellow), and 2000 ms (dark blue) ramp times to reach the 60 LPM peak flow rate corresponding to 20, 5.88, 2, 1, and 0.5 L/s^2^ acceleration rates, respectively, for a 4 L inspired volume. (**B**) The profiles show changes in inspired volumes at the 50 ms ramp time corresponding to a 20 L/s^2^ acceleration rate for the 1 L (red), 2 L (purple), and 4 L (blue) volume, and 2000 ms ramp time corresponding to a 0.5 L/s^2^ acceleration rate at the 1 L (green), 2 L (dark red), and 4 L (dark blue) volume to reach the 60 LPM peak flow rate. (**C**) The profiles show changes in inspired volume at a 25 ms ramp time corresponding to a 20 L/s^2^ acceleration rate at the 1 L (red), 2 L (purple), and 4 L (blue) volume and 1000 ms ramp time corresponding to a 0.5 L/s^2^ acceleration rate at the 1 L (green), 2 L (dark red), and 4 L (dark blue) volume to reach the 30 LPM peak flow rate.

**Figure 2 pharmaceutics-15-00934-f002:**
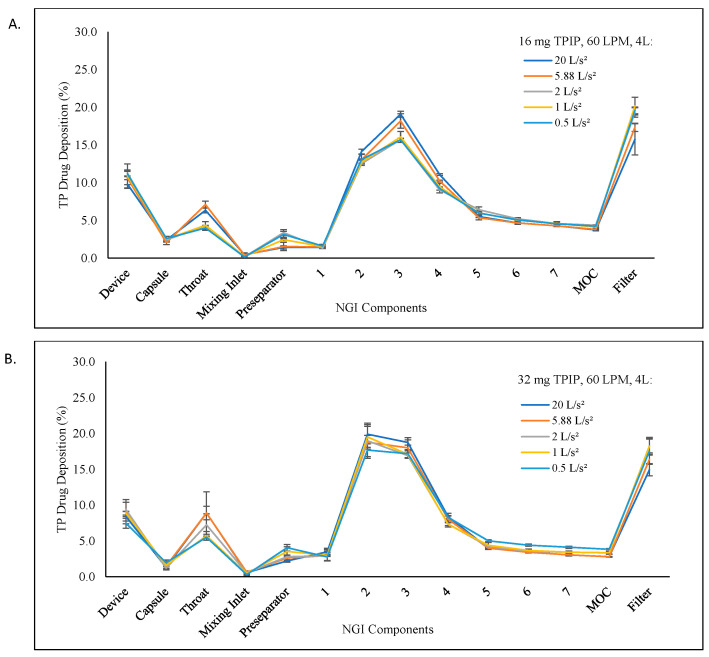
TP aerosol deposition profiles in the NGI as a function of acceleration rate. The profiles show the percentage of TP recovered (mean ± SD) from each component of the NGI at each inhalation acceleration rate to reach the 60 LPM peak inspiratory flow rate with a 4 L inhalation volume. (**A**) 16 mg TPIP capsules. (**B**) 32 mg TPIP capsules. Acceleration rates: 20 L/s^2^ (blue), 5.88 L/s^2^ (orange), 2 L/s^2^ (gray), 1 L/s^2^ (yellow), and 0.5 L/s^2^ (light blue).

**Figure 3 pharmaceutics-15-00934-f003:**
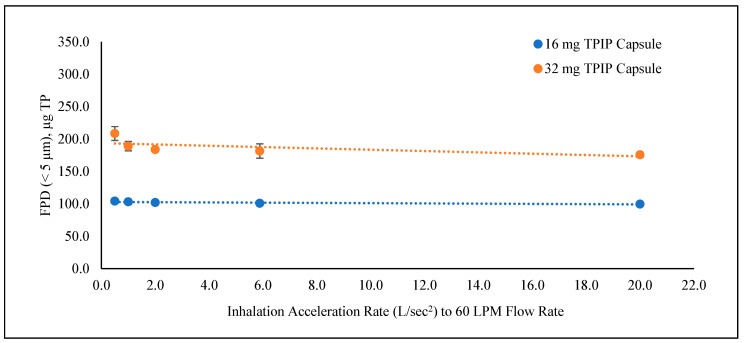
TP fine particle dose (FPD) as a function of inhalation acceleration rate to reach the 60 LPM peak inspiratory flow rate for a 4 L inhalation volume. The dotted lines show the change in FPD values for the 16 mg TPIP (blue) and 32 mg TPIP (orange) capsules as the acceleration rate increased from 0.5 L/s^2^ (2000 ms ramp) to 20 L/s^2^ (50 ms ramp).

**Figure 4 pharmaceutics-15-00934-f004:**
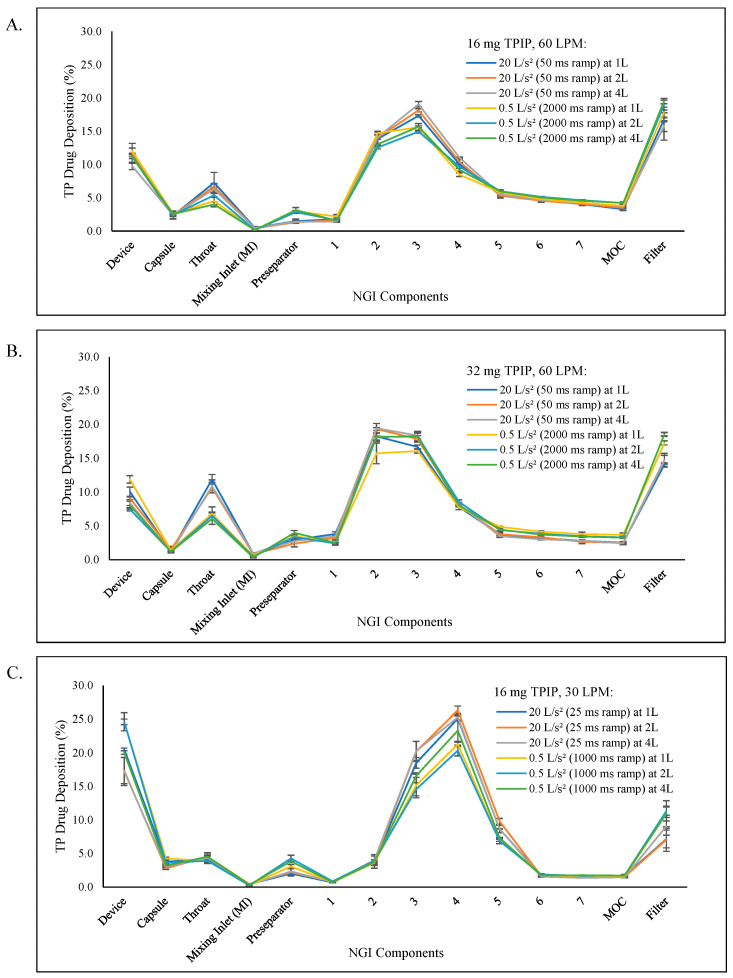
TP aerosol deposition profiles in the NGI as a function of inhaled volume. The profiles show the percentage of TP recovered (mean ± SD) from each component of the NGI for each inhalation volume with 20 and 0.5 L/s^2^ acceleration rates to reach the 60 and 30 LPM peak inspiratory flow rates. (**A**) 16 mg TPIP capsules at a 50 ms ramp time for the 1 L (blue), 2 L (orange), and 4 L (gray) volumes, and 2000 ms ramp time for the 1 L (yellow), 2 L (light blue), and 4 L (green) volumes to reach the 60 LPM peak flow rate. (**B**) 32 mg TPIP capsules at a 50 ms ramp time for the 1 L (blue), 2 L (orange), and 4 L (gray) volumes, and 2000 ms ramp time for the 1 L (yellow), 2 L (light blue), and 4 L (green) volumes to reach the 60 LPM peak flow rate. (**C**) 16 mg TPIP capsules at a 25 ms ramp time for the 1 L (blue), 2 L (orange), and 4 L (gray) volumes, and 1000 ms ramp time for the 1 L (yellow), 2 L (light blue), and 4 L (green) volumes to reach the 30 LPM peak flow rate.

**Figure 5 pharmaceutics-15-00934-f005:**
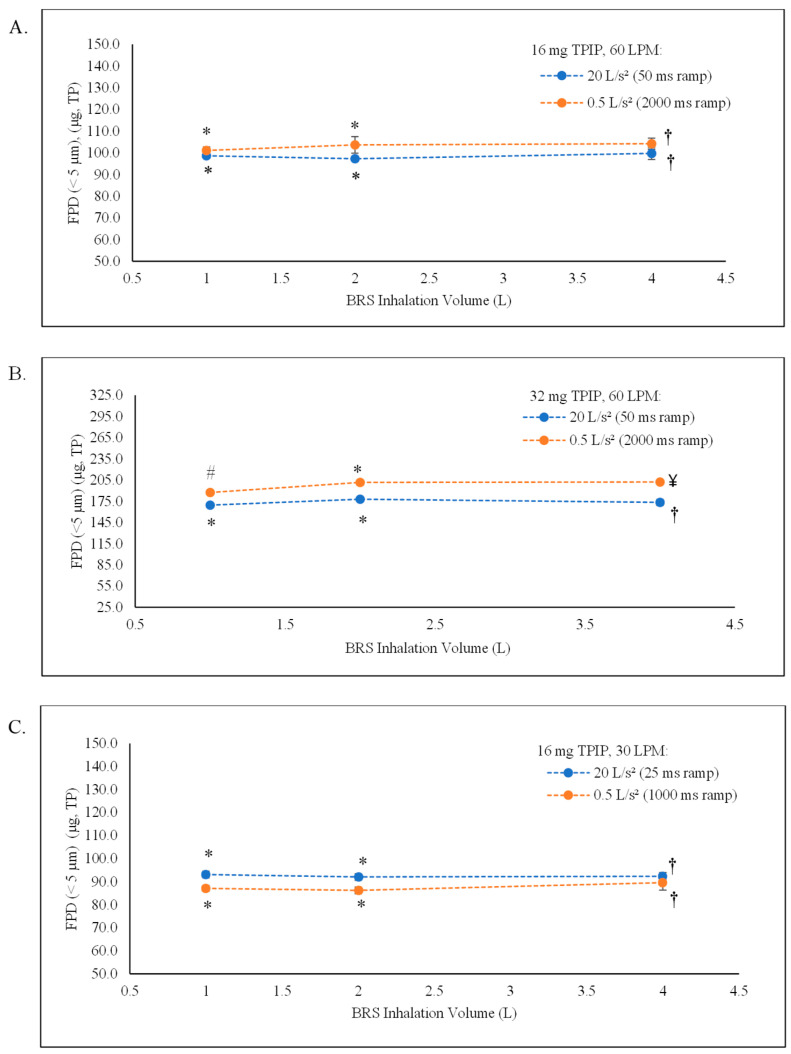
TP fine particle dose (FPD) trend lines as a function of the total inhalation volume. The dotted lines show the change in FPD values as inhalation volume increases for the 20 L/s^2^ (blue) and 0.5 L/s^2^ (orange) acceleration rates at the 30 and 60 LPM peak inspiratory flow rates. (**A**) 16 mg TPIP capsules at the 60 LPM flow rate. (**B**) 32 mg TPIP capsules at the 60 LPM flow rate. (**C**) 16 mg TPIP capsules at the 30 LPM flow rate. * *p*-value > 0.05 for the 1 and 2 L inhalation volumes compared to the 4 L inhalation volume, # *p*-value < 0.05 for the 1 L inhalation volume compared to the 4 L inhalation volume, † *p*-value > 0.05 for the overall comparison of inhalation volumes for the 20 L/s^2^ acceleration rate (16 mg and 32 mg) and 0.5 L/s^2^ acceleration rate (16 mg), and ¥ *p*-value < 0.05 overall comparison of inhalation volume for the 0.5 L/s^2^ acceleration rate (32 mg) associated with data from Table 3 and Table 4.

**Table 1 pharmaceutics-15-00934-t001:** Aerosol performance characteristics for the 16 and 32 mg TPIP capsules as a function of inhalation flow rate ramp times: 60 LPM peak inspiratory flow rate and 4 L inhalation volume. MMAD = mass median aerodynamic diameter; GSD = geometric standard deviation; FPF = fine particle fraction; FPD = fine particle dose; TP = treprostinil palmitil.

Inhalation Ramp Time	n	Emitted Dose		MMAD	GSD	FPF (<5 µm)		FPD (<5 µm)
		%	µg, TP	µm		% of Emitted Dose	% of Loaded Dose	µg, TP
16 mg capsule containing 160 µg TP								
50 ms	3	79.8 ± 1.1	127.68 ± 1.70	1.27 ± 0.08	3.54 ± 0.11	78.1 ± 1.1	62.3 ± 1.7	99.71 ± 2.76
170 ms	3	81.3 ± 1.2	130.11 ± 1.95	1.18 ± 0.03	3.71 ± 0.05	77.6 ± 1.1	63.1 ± 0.8	100.89 ± 1.30
500 ms	3	79.8 ± 0.9	127.69 ± 1.46	1.04 ± 0.03	3.83. ± 0.05	80.0 ± 0.4	63.8 ± 1.0	102.11 ± 1.64
1000 ms	3	80.3 ± 2.7	128.47 ± 4.24	1.05 ± 0.05	3.93 ± 0.10	80.4 ± 0.3	64.5 ± 2.4	103.23 ± 3.80
2000 ms	3	81.8 ± 0.6	130.88 ± 1.00	1.06 ± 0.02	3.97 ± 0.10	79.6 ± 1.4	65.1 ± 1.6	104.23 ± 2.57
32 mg capsule containing 320 µg TP								
50 ms	3	81.3 ± 1.7	260.10 ± 5.55	1.66 ± 0.07	4.30 ± 0.21	67.6 ± 1.3	54.9 ± 1.3	175.81 ± 4.14
170 ms	3	82.7 ± 1.3	264.76 ± 4.01	1.51 ± 0.02	4.35 ± 0.10	68.6 ± 3.3	56.8 ± 3.5	181.59 ± 11.15
500 ms	3	81.3 ± 0.3	260.04 ± 1.07	1.38 ± 0.10	4.59 ± 0.26	70.7 ± 1.6	57.5 ± 1.1	183.93 ± 3.48
1000 ms	3	82.9 ± 0.4	265.20 ± 1.12	1.40 ± 0.13	4.57 ±0.23	71.3 ± 2.4	59.1 ± 2.3	189.18 ± 7.19
2000 ms	3	89.4 ± 2.2	286.14 ± 7.08	1.33 ± 0.10	4.32 ± 0.09	72.9 ± 2.0	65.2 ± 3.3	208.75 ± 10.62

**Table 2 pharmaceutics-15-00934-t002:** Comparison of the FPDs among the various combinations of inhalation acceleration rates (50 to 2000 ms flow rate ramp) for the 16 mg and 32 mg TPIP capsules: 60 LPM peak inspiratory flow rate and 4 L inhalation volume. *p*-values in red signify statistically significant differences. FPD = fine particle dose.

Ramp Time	FPD (<5 µm) *t*-Test 2-Tailed *p*-Values				
	vs. 50 ms	vs. 170 ms	vs. 500 ms	vs. 1000 ms	vs. 2000 ms
16 mg capsule containing 160 µg TP					
50 ms	-	0.540	0.265	0.265	0.106
170 ms	0.540	-	0.370	0.371	0.115
500 ms	0.265	0.370	-	0.664	0.294
1000 ms	0.265	0.371	0.664	-	0.724
2000 ms	0.106	0.115	0.294	0.724	-
Overall *p*-value (ANOVA: single factor)	0.282				
32 mg capsule containing 320 µg TP					
50 ms	-	0.447	0.060	0.049	0.007
170 ms	0.447	-	0.747	0.378	0.038
500 ms	0.060	0.747	-	0.318	0.018
1000 ms	0.049	0.378	0.318	-	0.057
2000 ms	0.007	0.038	0.018	0.057	-
Overall *p*-value (ANOVA: single factor)	0.005				

**Table 3 pharmaceutics-15-00934-t003:** Aerosol performance characteristics for the 16 and 32 mg TPIP capsules as a function of inhalation volume (1, 2 and 4 L) and acceleration rate (50 and 2000 ms flow rate ramps) for the 60 LPM peak inspiratory flow rate. MMAD = mass median aerodynamic diameter; GSD = geometric standard deviation; FPF = fine particle fraction; FPD = fine particle dose; TP = treprostinil palmitil.

Inhalation Parameters: Ramp Time, Inhaled Volume	n	Emitted Dose		MMAD	GSD	FPF (<5 µm)		FPD (<5 µm)
		%	µg, TP	µm		% of Emitted Dose	% of Loaded Dose	µg, TP
16 mg capsule containing 160 µg TP at 60 LPM (50 and 2000 ramp times corresponding to 20 and 0.5 L/s^2^ acceleration rates, respectively)								
50 ms, 1 L	3	81.1 ± 0.6	129.81 ± 0.96	1.24 ± 0.02	3.80 ± 0.07	76.0 ± 1.5	61.7 ± 0.7	98.70 ± 1.19
50 ms, 2 L	3	79.1 ± 1.9	126.53 ± 3.07	1.28 ± 0.01	3.68 ± 0.13	76.9 ± 1.0	60.8 ± 0.7	97.28 ± 1.12
* 50 ms, 4 L	3	79.8 ± 1.1	127.68 ± 1.70	1.27 ± 0.08	3.54 ± 0.11	78.1 ± 1.1	62.3 ± 1.7	99.71 ± 2.76
2000 ms, 1 L	3	82.3 ± 1.2	131.71 ± 1.92	1.19 ± 0.05	4.16 ± 0.05	76.8 ± 1.0	63.2 ± 1.0	101.13 ± 1.61
2000 ms, 2 L	3	82.5 ± 2.3	131.94 ± 3.63	1.06 ± 0.03	3.95 ± 0.05	78.6 ± 0.7	64.8 ± 2.4	103.68 ± 3.81
* 2000 ms, 4 L	3	81.8 ± 0.6	130.88 ± 1.00	1.06 ± 0.02	3.97 ± 0.10	79.6 ± 1.4	65.1 ± 1.6	104.23 ± 2.57
32 mg capsule containing 320 µg TP at 60 LPM (50 and 2000 ramp times corresponding to 20 and 0.5 L/s^2^ acceleration rates, respectively)								
50 ms, 1 L	3	84.1 ± 0.5	269.01 ± 1.62	1.68 ± 0.03	4.51 ± 0.07	63.0 ± 1.0	53.0 ± 0.8	169.43 ± 2.45
50 ms, 2 L	3	84.7 ± 1.3	271.12 ± 3.99	1.65 ± 0.02	4.30 ± 0.14	65.6 ± 1.0	55.6 ± 0.9	177.94 ± 2.73
* 50 ms, 4 L	3	81.3 ± 1.7	260.10 ± 5.55	1.66 ± 0.07	4.30 ± 0.21	67.6 ± 1.3	54.9 ± 1.3	175.81 ± 4.14
50 ms, 4 L	3	83.9 ± 0.5	268.37 ± 1.53	1.69 ± 0.07	4.35 ± 0.08	64.6 ± 1.5	54.2 ± 1.3	173.38 ± 4.28
2000 ms, 1 L	3	81.2 ± 1.3	259.99 ± 4.08	1.27 ± 0.08	4.37 ± 0.16	72.1 ± 1.4	58.6 ± 0.3	187.40 ± 0.88
2000 ms, 2 L	3	86.8 ± 0.6	277.59 ± 2.00	1.34 ± 0.03	4.25 ± 0.10	72.6 ± 1.4	63.0 ± 1.1	201.61 ± 3.49
* 2000 ms, 4 L	3	89.4 ± 2.2	286.14 ± 7.08	1.33 ± 0.10	4.32 ± 0.09	72.9 ± 2.0	65.2 ± 3.3	208.75 ± 10.62
2000 ms, 4 L	3	87.6 ± 0.9	280.22 ± 2.81	1.35 ± 0.03	4.33 ± 0.01	72.2 ± 0.2	63.2 ± 0.6	202.20 ± 1.96

* Original data from the acceleration rate study for the 16 mg and 32 mg capsule. No significant differences (*p* > 0.05) in the ED, MMAD, FPF, and FPD between the original and repeat runs.

**Table 4 pharmaceutics-15-00934-t004:** Comparison of the FPDs between various combinations of inhalation volumes (1, 2 and 4 L) at the longest and shortest flow rate ramp time for the 16 mg and 32 mg TPIP capsules for the 60 LPM peak inspiratory flow rate. *p*-values in red signify statistically significant differences. FPD = fine particle dose.

Ramp Time, Volume	FPD (<5 µm) *t*-Test 2-Tailed *p*-Values					
	vs. 50 ms 1 L	vs. 50 ms 2 L	vs. 50 ms 4 L	vs. 2000 ms 1 L	vs. 2000 ms 2 L	vs. 2000 ms 4 L
16 mg capsule containing 160 µg TP at 60 LPM (50 and 2000 ramp times corresponding to 20 and 0.5 L/s^2^ acceleration rates, respectively)						
50 ms, 1 L	-	0.206	0.588	0.103	0.097	0.028
50 ms, 2 L	0.206	-	0.229	0.027	0.049	0.013
50 ms, 4 L	0.588	0.229	-	0.485	0.218	0.106
2000 ms, 1 L	0.103	0.027	0.485	-	0.345	0.151
2000 ms, 2 L	0.097	0.049	0.218	0.345	-	0.846
2000 ms, 4 L	0.028	0.013	0.106	0.151	0.846	-
(ANOVA: single factor)	0.336			0.409		
Overall *p*-value (ANOVA: single factor)	0.022					
32 mg capsule containing 320 µg TP at 60 LPM (50 and 2000 ramp times corresponding to 20 and 0.5 L/s^2^ acceleration rates, respectively)						
50 ms, 1 L	-	0.016	0.237	0.000	0.000	0.000
50 ms, 2 L	0.016	-	0.194	0.005	0.001	0.000
50 ms, 4 L	0.237	0.194	-	0.005	0.001	0.000
2000 ms, 1 L	0.000	0.005	0.005	-	0.002	0.000
2000 ms, 2 L	0.000	0.001	0.001	0.002	-	0.811
2000 ms, 4 L	0.000	0.000	0.000	0.000	0.811	-
*p*-value (ANOVA: single factor)	0.050			0.004		
Overall *p*-value (ANOVA: single factor)	0.000					

**Table 5 pharmaceutics-15-00934-t005:** Aerosol performance characteristics for the 16 mg TPIP capsules as a function of inhalation volume (1, 2, and 4 L) and acceleration rate (25 and 1000 ms flow rate ramps) for the 30 LPM peak inspiratory flow rate. MMAD = mass median aerodynamic diameter; GSD = geometric standard deviation; FPF = fine particle fraction; FPD = fine particle dose; TP = treprostinil palmitil.

Inhalation Parameters: Ramp Time, Inhaled Volume	n	Emitted Dose		MMAD	GSD	FPF (<5 µm)		FPD (<5 µm)
		%	µg, TP	µm		% of Emitted Dose	% of Loaded Dose	µg, TP
16 mg capsule containing 160 µg TP at 30 LPM (25 and 1000 ramp times corresponding to 20 and 0.5 L/s^2^ acceleration rates, respectively)								
25 ms, 1 L	3	76.4 ± 0.7	122.30 ± 1.04	2.19 ± 0.11	2.34 ± 0.05	76.2 ± 1.7	58.2 ± 0.9	93.13 ± 1.45
25 ms, 2 L	3	76.1 ± 2.3	121.79 ± 3.61	2.25 ± 0.08	2.31 ± 0.10	75.6 ± 1.4	57.5 ± 0.9	92.08 ± 1.40
25 ms, 4 L	3	76.1 ± 1.8	121.83 ± 2.95	2.16 ± 0.08	2.40 ± 0.06	75.9 ± 1.8	57.7 ± 1.0	92.39 ± 1.60
1000 ms, 1 L	3	71.7 ± 1.8	114.73 ± 2.83	1.90 ± 0.11	2.58 ± 0.07	76.0 ± 1.3	54.5 ± 0.8	87.16 ± 1.27
1000 ms, 2 L	3	72.7 ± 1.1	116.30 ± 1.71	1.88 ± 0.08	2.67 ± 0.03	74.2 ± 1.1	53.9 ± 0.9	86.26 ± 1.44
1000 ms, 4 L	3	74.8 ± 2.3	119.67 ± 3.75	1.95 ± 0.11	2.54 ± 0.09	74.9 ± 0.6	56.0 ± 2.1	89.61 ± 3.29

**Table 6 pharmaceutics-15-00934-t006:** Comparison of the FPDs between various combinations of inhalation volumes (1, 2 and 4 L) at the longest and shortest flow rate ramp time for the 16 mg TPIP capsule: 30 LPM peak inspiratory flow rate. *p*-values in red signify statistically significant differences. FPD = fine particle dose.

Ramp Time, Volume	FPD (< 5 µm) *t*-Test 2-Tailed *p*-Values					
	vs. 25 ms 1 L	vs. 25 ms 2 L	vs. 25 ms 4 L	vs. 1000 ms 1 L	vs. 1000 ms 2 L	vs. 1000 ms 4 L
16 mg capsule containing 160 µg TP at 30 LPM (25 and 1000 ramp times corresponding to 20 and 0.5 L/s^2^ acceleration rates, respectively)						
25 ms, 1 L	-	0.415	0.581	0.006	0.004	0.165
25 ms, 2 L	0.415	-	0.813	0.011	0.007	0.298
25 ms, 4 L	0.581	0.813	-	0.011	0.008	0.259
1000 ms, 1 L	0.006	0.011	0.011	-	0.463	0.296
1000 ms, 2 L	0.004	0.007	0.008	0.463	-	0.182
1000 ms, 4 L	0.165	0.298	0.259	0.296	0.182	-
(ANOVA: single factor)	0.686			0.235		
Overall *p*-value (ANOVA: single factor)	0.003					

## Data Availability

Data is contained within the article or Appendix A.

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
