# Peer review of "Effect of Inhalation Profile on Delivery of Treprostinil Palmitil Inhalation Powder"

_pharmaceutics, 2023, doi:10.3390/pharmaceutics15030934_

Round 1

Reviewer 1 Report

The present article discusses the technical performance and assessment of Treprostinil palmitil (TP) as an inhalable powder for use with a DPI. The methods and data presented seem sound and the conclusions support that the efficacy of the delivery is likely to be independent of various patient-specific inhalation profiles, which is good news. While the originality of the work is not high, the results are very tangible and useful for pulmonary drug delivery endpoints.

Minor point: Table 3 is slightly overwhelming and could be included in the SM instead.

Author Response

Thank you for the positive comments. The authors agree that while there is not a high level of innovation, the data adds value to development scientists who need to consider such studies. Very few in vitro studies of inhaler performance under realistic use scenarios are present in the literature.

Both Tables 1 and 3 have been abridged for the manuscript and the additional content is now available in the supplementary materials. The bottom third of the content in Table 3 has been moved to a new Table 5.

Reviewer 2 Report

The article is clearly written and the experimental part is very robust. I suggest accepting the text and evaluating the following suggestions:

-the abstract is too extended. I suggest condensing it by removing the experimental details and focusing more on the key results of the research

-line 55. I suggest specifying the dosage of Tyvaso 

-Fig 2: it is not clear to me what the "filter" inserted downstream of the MOC is, is it a component external to the NGI? Shouldn't the MOC capture all particles smaller than 0.34 or 0.54 µm?

I don't think it's mentioned in the paragraph 2.3

-Table 1 : Given the small value of MMAD and the large deposition in the final part of the impactor, I suggest the authors to calculate also the Extrafine dose < 2µm

-Table 1: a small discussion / explanation about the lung portion where the TP has its target would be useful to understand if the peripheral deposition profile would be advantageous

-Table 3: do not insert the “% of loaded dose” column under the FPD cell. it is a percentage fraction and not a dose, it would be more correct to insert it under FPF%

-Line 462, Tobi has 112mg of API embedded into 200 mg of powder which is all to be inhaled. Two inhalations are required for the 4 capsules.

-Line 477-line 498: I would like to point out to the authors that, although it may be a limitation that the reported data were collected in vitro, a robust correlation has been reported between the emitted dose (of a carrier free powder emitted by an RS01 device) obtained from healthy volunteers and in vitro at 1,2,3 L of air volume at 4kPa (Buttini, F. et al.Dose administration maneuvers and patient care in tobramycin dry powder inhalation therapy. IJP 548, 182–191 (2018)

I believe these data may be predictive of the dose extracted from patients. 

On the other hand, it is curious that the volume of air passed does not have a particular impact on the emitted dose, which was observed for consecutive inhalations through a 120 mg capsule of tobramycin.

I believe this may be because 32 or 16 mg of powder can be extracted from a single inhalation act where even 1L is able to move and extract the entire powder bed. 

I leave to the authors the freedom to evaluate whether these considerations may be of interest to the discussion

Author Response

Thank you for the positive comments.

-the abstract is too extended. I suggest condensing it by removing the experimental details and focusing more on the key results of the research

The authors agree and have reduced the abstract as you have suggested.

-line 55. I suggest specifying the dosage of Tyvaso 

We have included the dosage for the Tyvaso DPI (16, 32, 48 and 64 ug) which is a relevant comparator.

-Fig 2: it is not clear to me what the "filter" inserted downstream of the MOC is, is it a component external to the NGI? Shouldn't the MOC capture all particles smaller than 0.34 or 0.54 µm?

I don't think it's mentioned in the paragraph 2.3

The text in 2.3 has been updated. The filter is a component external to the NGI. The manuscript describing the original development of the NGI (https://doi.org/10.1089/089426803769017659) states that a filter may be necessary to capture very fine particles: “Some samples may contain particles that are so fine that they are not collected by the MOC. However, since the particle size distributions generated by most inhalers are larger than the cut size of the MOC, its efficiency is satisfactory for the majority of drug product formulations. Nevertheless, its effectiveness must be evaluated for any new formulation, or inhaler device, during method development by placing a filter down-stream of the MOC and determining the magnitude of the portion of the dose (if any), that penetrates the MOC.”

-Table 1: Given the small value of MMAD and the large deposition in the final part of the impactor, I suggest the authors to calculate also the Extrafine dose < 2µm

While there is not at present a unified definition for extrafine dose, this parameter has been calculated and added to the two supplemental tables that refer to the data reported in Tables 1 and 3.

-Table 1: a small discussion / explanation about the lung portion where the TP has its target would be useful to understand if the peripheral deposition profile would be advantageous

This is an excellent question regarding inhaled TP. The optimal location in the lung for delivery of TP may be slightly more peripheral than for delivery of TRE. That is because the conversion of TP to TRE occurs over a longer time frame, and it is possible that some of the TP may have been swept up by the mucociliary escalator into more central regions of the lung prior to conversion to TRE. Unfortunately, even though inhaled TRE has been on the market for almost two decades, there is limited discussion in the literature as to its optimal regional target in the lung. We don’t want to speculate on this as the clinical trial data will speak for itself. However, we have included some text in the discussion section.

-Table 3: do not insert the “% of loaded dose” column under the FPD cell. it is a percentage fraction and not a dose, it would be more correct to insert it under FPF%

The authors have reformatted the table and moved the FPD (% of loaded dose) under FPF (% of loaded dose).

-Line 462, Tobi has 112mg of API embedded into 200 mg of powder which is all to be inhaled. Two inhalations are required for the 4 capsules.

The text has been clarified.

-Line 477-line 498: I would like to point out to the authors that, although it may be a limitation that the reported data were collected in vitro, a robust correlation has been reported between the emitted dose (of a carrier free powder emitted by an RS01 device) obtained from healthy volunteers and in vitro at 1,2,3 L of air volume at 4kPa (Buttini, F. et al.Dose administration maneuvers and patient care in tobramycin dry powder inhalation therapy. IJP 548, 182–191 (2018). I believe these data may be predictive of the dose extracted from patients. 

Thank you. We have cited that manuscript in the discussion.

On the other hand, it is curious that the volume of air passed does not have a particular impact on the emitted dose, which was observed for consecutive inhalations through a 120 mg capsule of tobramycin. I believe this may be because 32 or 16 mg of powder can be extracted from a single inhalation act where even 1L is able to move and extract the entire powder bed. I leave to the authors the freedom to evaluate whether these considerations may be of interest to the discussion.

We agree with your assessment. The take-home message is that 1 L of inspired air is adequate to fully deaggregate a dispersible powder like TPIP that is presented in relatively small capsule loads of 16 and 32 mg compared to TOBI Podhaler.

Reviewer 3 Report

The authors investigated the effect of inhalation profile on delivery of Treprostinil palmitil inhalation powder. The study is interesting and the experiments were well designed. The manuscript can be accepted after addressing the following minor points:

1- Line 21, LPM "full name should be provided at its first appearance"

2- The Abstract is too long. It should be summarize and be concise to disclose the important findings.

3- The manuscript should be checked for typographical mistakes.

Author Response

Thank you for the positive comments.

LPM has been defined at its first appearance.

The abstract text has been reduced.

We will robustly review the final typeset manuscript for typos.